# Context-Aware Content Selection and Message Generation for Collective Perception Services

**Ameni Chtourou, Pierre Merdrignac *** 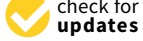 **and Oyunchimeg Shagdar ***

REVECOM TEAM, VEDECOM Institute, 23 bis Allee des Marronniers, 78000 Versailles, France;
ameni.chtourou@vedecom.fr
* Correspondence: pierre.merdrignac@vedecom.fr (P.M.); oyunchimeg.shagdar@vedecom.fr (O.S.)

**Abstract:** Collective perception messages (CPMs) inform neighbors about detected objects and their current status. Although CPMs can greatly contribute to improving vehicle perception, their performance will degrade if the wireless channel is saturated and/or vehicles must process excessively high amount of data to build such a perception. Channel saturation and processing overhead can be induced by transmissions of CPMs if vehicles and roadside infrastructure transmit CPMs too often with too much redundant information. Indeed, according to the current draft standard, the CPM generation frequency at individual vehicles and infrastructure can be as high as 10 Hz and depending on the perception capability, individual CPMs can be quite heavy (around 1500 Bytes), containing redundant information. It is hence important to integrate techniques that reduce information redundancy while providing sufficient perception with optimal resource use. In this paper, we propose context-aware communication schemes that control CPM content selection and transmission. Two types of contexts, particularly radio resource use (channel busy ratio: CBR) and infrastructure availability, have been considered in the proposed schemes. Using a network simulator, we evaluated the performances of the proposed schemes in terms of CBR, packet delivery ratio and awareness ratio. The simulation results show that the scheme that takes into account both resource use and infrastructure availability offers the best performance on all the above metrics.

**Keywords:** C-ITS; collective perception; context awareness; communication control; simulations

## 1. Introduction

Automated driving is possible only if the vehicle has a complete situation awareness, i.e., perception of their environment. Initially, vehicles collect measurements via on-board sensors, e.g., radar, LiDAR, cameras that enable advanced driving assistance systems (ADAS), offering enhanced services in terms of comfort and safety to the passengers. However, such a perception based only on local observations is not enough for complex driving maneuvers, e.g., lane changing in urban areas or lane merging on highway, due to sensor inaccuracy or limited field of view. Such a local perception can be extended by cooperation between traffic entities, i.e., vehicles, roadside infrastructure, traffic management centers, through information exchange in so-called Cooperative Intelligent Transportation Systems (C-ITS). Among applications enabled by C-ITS, augmented perception relies on multiple messages, i.e., Cooperative Awareness Message (CAM) [1] and Collective Perception Message (CPM) [2], to complement the local perception and assess state variables of other users. Although CAMs are mainly generated by vehicles, CPMs can be generated at vehicles and roadside infrastructure equipped with on-board sensors.

Currently, ETSI is studying CPM to specify a collective perception service (CPS) including a message format and generation rules. ETSI analysis showed that average number of objects per CPM can be very low while updates are frequent (10 messages per second) [2] which leads to an unnecessary increase of channel load. In addition, authors of [3,4], confirmed that generation of CPM is frequent ant that CPM tend to be small

messages reporting few objects. Hence, a heavy usage of communication resources is induced and, eventually, V2X communication reliability are degraded due to loss of some messages. Consequently, it is necessary to design advanced techniques that dynamically control CPM redundancy on the wireless channel while ensuring a higher awareness level of the driving environment.

Because applications of C-ITS have to operate in heterogeneous conditions, in terms of environmental situations (e.g., road types, weather conditions) and technologies (e.g., communication media, network infrastructure), context awareness [5] can be considered to be a key aspect to control protocols and services, such as CPS. Indeed, for connected vehicles, context has been defined by Sepulcre et al. [6] as "*a collection of measured, exchanged and inferred knowledge that characterizes the vehicular environment and the communication needs and conditions of a vehicular node*", which can then be used for reasoning and communication control. In addition, the authors of [6] proposed a context-aware communications system architecture, illustrated in Figure 1, in which the management layer of the ETSI/ISO C-ITS reference architecture [7] implements context awareness through the following three phases.

- Context acquisition: which consists of collecting data from different sources including V2X communication, sensors and GNSS.
- Context reasoning: that aims to employ intelligent techniques to obtain a higher level of contextual data.
- System control: which provides users with services depending on their actual situations, i.e., in charge of making appropriate decisions for every layer such as technology selection or message generation frequency optimization.

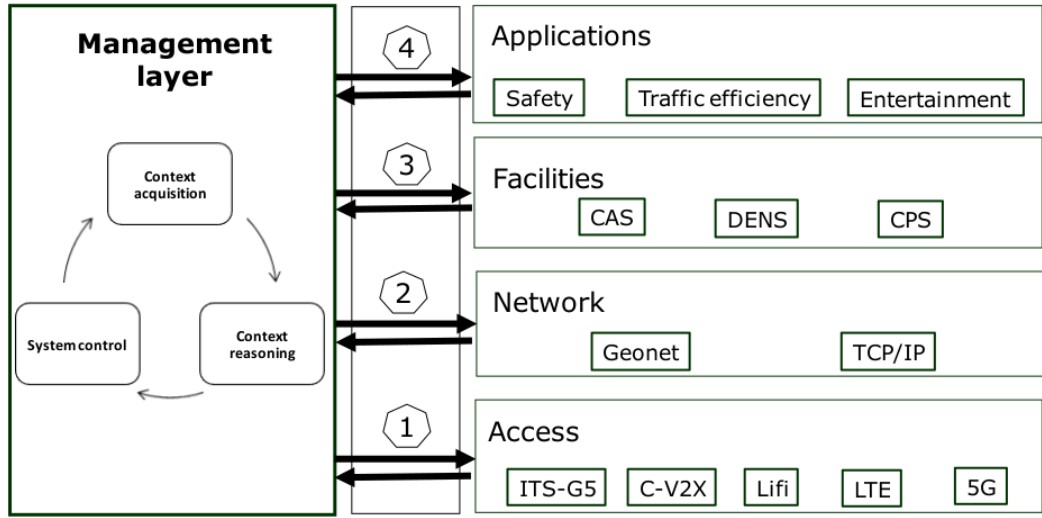

**Figure 1.** Context-aware communication architecture.

Although many contexts can be imagined, we particularly believe that for road safety applications, at least C-ITS applications' requirements, radio resource availability, and environmental context, shall be considered (see Figure 2). A collision warning application requirement can be calculated using the ego-vehicle position, its kinematic status, the current road layout, and etc. [8]. The radio resource availability can be measured by e.g., channel busy ratio (CBR). Finally, as environmental condition, one may consider the presence of roadside communication infrastructure e.g., roadside unit (RSU) and/or mobile edge computing (MEC) devices that may be equipped with sensors.

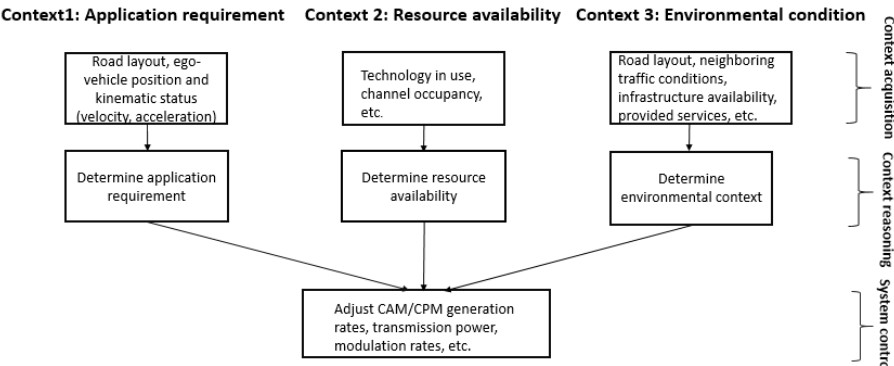

**Figure 2.** Context awareness for collision avoidance applications.

In communication systems, it is not new to take into account radio resource availability. Indeed, ETSI has defined a distributed congestion control (DCC) framework to allow communication protocols adapting their parameters based on the radio resource availability, often via the metric channel busy ratio (CBR). Multiple DCC algorithms have been proposed to control facilities layer message generation rate, packet forwarding parameters , transmission power and modulation rate etc. [9–12]. In our previous work [8], we showed that DCC algorithms that takes into account only resource availability do not meet the application requirement, demonstrating a need for considering more than one context to control a communication system. In [13], we presented benefits of considering presence of roadside infrastructure, which is an environmental context, for collective perception service. Indeed, roadside infrastructure is often installed in such a way that it has higher communication and perception coverage than vehicles, and it has higher computing capability. To this end, in this paper, we are interested in designing and developing CPM content selection algorithms that build cooperative perception while avoiding channel congestion. The key contributions of the current work are as follows.

- Design and develop algorithms to control CPM generation and content selection based on channel status.
- Design and develop algorithms to control CPM generation and content selection based on infrastructure availability.
- Evaluate proposed algorithms in terms of CBR, PDR and awareness rate.

The rest of this paper is organized as follows: Section 2 reviews existing works about CPM generation rules and redundancy mitigation techniques. In Section 3, we propose different algorithms that aim to control CPM generation and content selection based on channel status, CBR calculation and infrastructure availability. In Section 4, we evaluate the performance of previously described algorithms and compare them to default settings. Finally, we conclude our work in Section 5.

## 2. Related Works

Among various cooperative driving applications, cooperative perception is considered to be an emerging technology for improving road safety with connected and automated vehicles (CAV) [14]. Cooperative perception means information-based cooperation by sharing vehicle-embedded sensors data via V2X Communications and has shown its potential to extend perception range of CAV [15–18].

As sharing raw sensor data [19] and statistical environment representations [20] have been considered to enable cooperative perception, recent works tend to rely on representation of high-level features (e.g., object position, speed, size, class,...) encoded in dedicated messages such as the *environmental perception message* introduced in [21], later harmonized by ETSI in the so-called Collective Perception Message [2].

Similar to CAMs, CPMs tend to be radio resource-consuming [21]. To limit potential channel congestion, which is one main issue for the 5.9 GHz spectrum used by C-ITS, ETSI has specified distributed congestion control (DCC) framework [12] and suggested differ-

ent DCC mechanisms [22,23] to control communication parameters (message generation frequency, transmission power, etc.) based on the current channel load expressed by CBR. Several efforts have been made to establish message generation rules and to determine the information that should or should not be included in CPMs based on channel load status [2,3,24]. However, in [25], the authors show that DCC can have negative effects on awareness level and latency of cooperative perception. Moreover, DCC does not consider any information related to messages content, thus, it cannot establish a priority between CPMs, for example, to favor A CPM that contains an occluded vehicle over a CPM with already detected vehicles.

In addition, some studies demonstrated significant information redundancy in CPMs transmitted by different vehicles due to their overlapping sensors field of view (FOV) [13,26,27]. Such a situation may load the communication channel with unnecessary redundant information, potentially leading to channel congestion, thus reducing the service reliability. Hence, efforts have been made to mitigate redundancy by selecting CPM content, i.e., choosing objects that are relevant to be included in a transmitted message [28,29].

In its technical report TR 103 562 [2], ETSI studied redundancy mitigation technique and suggested some rules based on frequency, dynamics, confidence, entropy, object self-announcement and distance. Although no clear conclusion on which of such rules should be implemented, the report highlights advantages and drawbacks of each of these approaches and leaves the opportunity for further research. To complement this study, Table 1 gives an overview of some approaches of CPM data content selection.

First, redundancy control is the main objective of multiple approaches, relying either on detected objects dynamics [3,29] or on probabilistic modelling of sensor field of view (FOV) [28] to decide whether an object detected by on-board sensors should be included in a CPM. The authors of [29] have shown that redundancy can be reduced, up to 4 times, while maintaining high awareness rate for short V2X communication range, i.e., below 100 m. Complementary, the authors of [28] showed up to 6.5 times of redundancy can be observed when the penetration rate of CAV is 50%. Additionally, this work introduces a redundancy control algorithm that randomly selects objects in CPM based on the transmission probability. Results show that a high awareness ratio is maintained while significantly reducing redundancy. However, such approach may be unreliable in the case a critical object is not shared in the V2X network due to the random data selection process.

Secondly, the authors of [30] designed a *value-anticipating cooperative framework* which assesses a *value of information* measurement for every detected object. Here, *value of information* is defined as the relative entropy between an estimated probability distribution function (pdf) of object state by the ego vehicle and a predicted pdf of object state for a remote vehicle. Then, a high *value of information* means that an object is of interest for a remote vehicle and should be included in the CPM. Although redundancy mitigation is not the primary objective in this work, such data content selection by anticipating the need to transmit a data object is a possible approach as suggested in [2].

Finally, channel overloading being one major issue with CPM dissemination, it has been shown that controlling CPM content can have a positive impact on channel usage [24,31]. In [32], the authors proposed to assess channel congestion to adapt object inclusion policies. However, none of these works investigate redundancy control among CPMs transmitted by multiple stations.

As a conclusion, the emergence of cooperative perception to support automated driving has led to a higher demand in terms of data transfer between road users. Due to the limited capacity of any communication media, new approaches are needed to ensure an efficient usage of available communication resources while maintaining high reliability for vehicular applications. Therefore, intelligent approaches for selection of CPM content are needed to enable a large deployment of cooperative perception. Current approaches seek to minimize data redundancy when sending CPMs. However, such redundancy can also have positive effects, for example, to improve accuracy or security [33–35]. In addition, approaches for channel load control do not take in account message content to prioritize

the transmitted packets and few works considered channel load as a metric for CPM content selection.

**Table 1.** Existing CPM data content selection approaches.

| Ref | Principle | Approach | Redundancy Control | Channel Load |
|-----|-----------|----------|--------------------|--------------|
| [24] | Periodic verification of conditions on detected object (Novelty, Distance & Speed Variation, Age) | Frequency and Dynamics-based | No | Yes, for algorithm evaluation |
| [31] | Vehicles equipped with sensor can extend their CAM to share characteristics ((position, speed,...) on behalf of their neighbors | Object self-announcement | No | Yes, for algorithm evaluation |
| [3,29] | Consider received objects in addition to the detected objects when applying dynamics-based generation rules from [2] | Dynamics-based | Yes | No |
| [28] | Analyze data redundancy from Line-Of-Sight (LOS) model and estimation of vehicles density and objects selection from probability assignment function | Confidence & Distance | Yes | No |
| [30] | Anticipate the value of information, i.e., added knowledge of sharing a detected object from an ego vehicle to one of its neighbors | Entropy | No | No |
| [32] | Assess congestion to select between different object inclusion policies ((1) include all objects, (2) include objects satisfying dynamics-based generation rules and randomly include other objects (3) include only objects selected from generation rules) | Dynamics-based with DCC | No | Yes for transmission control |
| This work | Context-aware CPM generation and content selection based on channel and roadside infrastructure availability | Dynamics-based | yes | yes |

Therefore, in our work, we introduce a context-aware CPM content selection and transmission control that assesses two factors (1) channel busy ratio as communication resource availability factor and (2) roadside infrastructure availability as environmental conditions factor. Three schemes have been proposed that differ in terms of context use and content selection algorithm.

## 3. Context-Aware CPM Content Selection Algorithms

In this section, we propose several CPM content selection schemes that take into account radio resource availability and/or presence of roadside infrastructure. The target CPM content selection system architecture for a vehicle is depicted in Figure 3. It should be noted that the architecture may be applied to roadside infrastructure providing CP service. Nevertheless, roadside infrastructure has higher communication, perception, and computing capability and lower density compared to vehicles, we consider that a CPM content selection control should be applied particularity to vehicles that tend to overload

wireless channel whereas their contribution for cooperative perception is lower than that of roadside infrastructure [13].

As can be seen in the figure, a vehicle has local perception module that processes data from the on-board sensors and creates lists of objects perceived by the sensors. The vehicle also has *collective perception service (CPS)* consisting of a client module (*CPS client*), which processes received CPMs, and a server module (*CPS server*), which is responsible for generating and transmitting CPMs. *Data fusion module* fuses the data obtained from the local perception module and the CPS client, and maintains *extended perception table*. In particular, the table consists of five columns: (A)–(E) (see Figure 3) that are:

(A)　　Object ID: an identification provided by the data fusion module,
(B)　　Object description: describes the object, particularly perceived time, position, velocity, type, dimension, etc.,
(C)　　Perceived by local sensors: indicates if the object is detected by the ego vehicle,
(D)　　Perceived by neighboring vehicles: the number of neighboring vehicles that detected the object,
(E)　　Perceived by RSUs: the amount of roadside infrastructure that detected the object.

Finally, upon reception of a list of objects detected by the *local perception module*, CPM content selection module fills outgoing CPMs and provides the *CPS server* for a transmission. Although it may include all the objects perceived by the local sensors (*Default* scheme), in this section, we proposed different schemes, as listed in Table 2, that may select contents (list of objects) for inclusion in outgoing CPMs by taking into account radio resource use (CBR) and/or infrastructure availability. The resource availability, is measured by a *PHY layer channel monitoring module* using the channel busy ratio (CBR) metric.

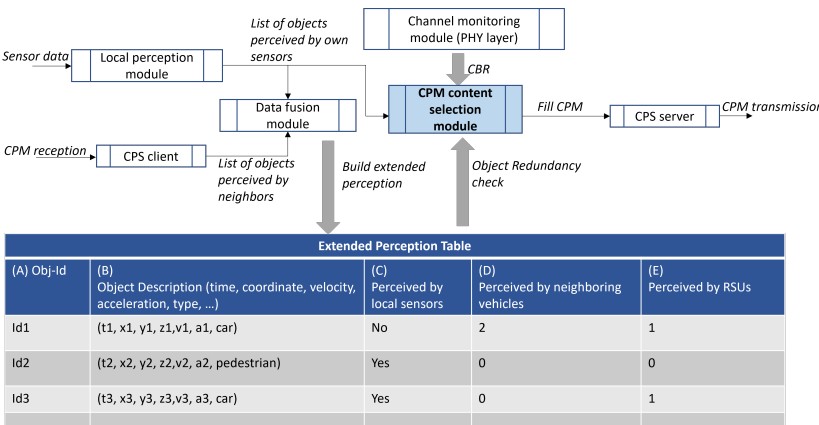

**Figure 3.** System diagram for Context-aware CPM generation and content selection.

As mentioned earlier, the *Default* scheme is not context-aware, in other words, CPMs are periodically broadcasted containing all the objects detected by its own sensors whether channel is congested or neighboring vehicles/RSUs have already detected and informed the object(s) via CPMs. The below subsections describe the remaining context-aware schemes.

**Table 2.** CPM generation and content selection algorithms.

| Scheme | Content Selection | CBR Consideration | Infra Consideration |
|---|---|---|---|
| *Default* | No | No | No |
| *CBR-Binary* | No | Yes | No |
| *CBR-Selective* | Yes | Yes | No |
| *CBR & Infra selective* | Yes | Yes | Yes |

### 3.1. CBR-Binary Scheme

The *CBR-binary* scheme transmits a CPM if only if the number of objects that are uniquely detected by the ego vehicle (i.e., none of the neighboring vehicles/RSU detected the same object) exceeds a threshold, *ThresholdBinary*, which is determined by CBR. Specifically, upon reception of a list of objects, $A = [a_i]$ perceived by own sensors (from the *local perception module*), the *CPM content selection module* determines if it shall send a CPM containing those objects (by the *CPS server*, see Figure 3).

As can be described in Algorithm 1, for each object, the *CPM content selection module* verifies, by referring to the columns (D) and (E) of the *extended perception table*, if any of its neighbors (vehicle or roadside infrastructure) has detected the same object. Once the above verification procedure has been finalized, the *CPM content selection module* compares the number of objects that are detected only by ego vehicle against a threshold value *ThresholdBinary*. If the number of such objects does not exceed *ThresholdBinary*, no CPM will be transmitted. Otherwise, the *CPM content selection module* fills a CPM with the objects in the *A* list and provides it to the *CPS server* for a transmission.

---

**Algorithm 1:** *CBR-binary* scheme.

```
/* A CPM is transmitted if only if the number of objects that are
   uniquely detected by the ego vehicle (i.e., none of the
   neighboring vehicles/RSU detected the same object) exceeds a
   threshold value, ThresholdBinary.                            */
```
**Initialize:** ThresholdControl(*Init*, $InitThr_{binary}$, $\delta_{binary}$)
**Input** : $A = [a_i], i = 1, 2, \cdots, n,$    `// a list of objects (more precisely,`
        `object IDs), perceived by the local perception module`
*shallSend*=CbrBinary(*A*)
**if** *ShallSend == True* **then**
    │   *Fill a CPM with the objects in A and provide the CPS server module for a*
    │   *transmission.*
**end**
**Function** CbrBinary(*A*):
    │ *ThresholdBinary*=ThresholdControl(*GetThr*,0,0)
    │ *count* = 0
    │ *shallSend* = *True*
    │ **for** $i \leftarrow 0$ **to** $n - 1$ **do**
    │    **if** $Column\_D(a_i) > 0$   *or*   $Column\_E(a_i) > 0$ **then**
    │     │   *count++*          `// the object is perceived by a neighbor.`
    │    **end**
    │ **end**
    │ **if** $size(A) - count \leq ThresholdBinary$ **then**
    │    │ *shallSend = False*
    │ **end**
    │ **return** *shallSend*
**End Function**

---

*ThresholdBinary* is adjusted by the *CPM content selection module* each time when a new CBR value is notified by the *PHY layer channel monitoring module*. The threshold control algorithm is depicted in Algorithm 2. In fact, the same algorithm is used for all the schemes that consider CBR i.e., *CBR-binary*, *CBR-selective*, and *CBR & Infra selective*. As can be seen in Algorithm 2, the threshold control algorithm has two parameters *Threshold* and a step parameter $\delta$. The *Threshold* parameter is incremented or decremented by *delta* each time a new CBR value is available. Therefore, depending on the $\delta$ value, *Threshold* can be modified quickly or slowly. It is up to each scheme to define the initial value of *Threshold* and the value of $\delta$. Nevertheless, we believe that the initial value of *Threshold* shall be configured such that the system operates similar to the *Default* algorithm in the initial condition. The step parameter, $\delta$, on the other hand, shall be chosen carefully to ensure that the system is stable yet can quickly adapt to the dynamics of the wireless channel.

As can be seen in Algorithm 1, *CBR-binary* sets the initial value of *Threshold* and the step parameter $\delta$ to *InitialThreshold$_{binary}$* and *delta$_{binary}$*, respectively. As can be seen in Algorithm 2, the threshold control algorithm increments *Threshold* by $\delta$, if the current CBR exceeds a predefined maximum CBR threshold, $CBR_{max}$ (channel is congested).

Finally, it should be noted that the scheme decides whether to transmit the complete list of detected objects. Therefore, the scheme controls the transmission opportunity (the transmission interval) of CPMs but not CPM contents.

---

**Algorithm 2:** *Threshold Control* function.

**Input** : *CBR*
**Output**: *Threshold*
```
/* Each time, when a new CBR value is available, the Threshold value
   is adapted.                                                     */
```
**if** *a new CBR value is available* **then**
 **if** $CBR < CBR_{min}$ **then**
  *Threshold* = max(0, *Threshold-$\delta$*) // the channel is sparse, reduce
  the threshold.
 **else if** $CBR > CBR_{max}$ **then**
  *Threshold* += $\delta$  // the channel is congested, increase the
  threshold.
**end**
```
/* Interacts with external functions (CBR − Binary or CBR − Selective)
   either to initialize Threshold, and the step parameter, δ or to
   provide the current Threshold value.                            */
```
**Function** ThresholdControl(*Key,Thr,ValDelta*):
 **if** *Key = "Init"* **then**
  *Threshold = Thr*
  $\delta$=*ValDelta*
 **else if** *Key = "GetThr"* **then**
  **return** *Threshold*
**End Function**

---

On the other hand, *Threshold* is decremented by $\delta$, if the CBR value is below a predefined minimum CBR threshold, $CBR_{min}$ (i.e., the channel is sparse). If the CBR is in the range of [$CBR_{min}$, $CBR_{max}$], *Threshold* is kept unchanged. Therefore, as it is described in Algorithm 1, if the channel is sparse, the *ThresholdBinary* takes on a lower value and hence the *CBR-binary* scheme tends to transmit CPMs even it contains few uniquely detected objects. On the contrary, if the channel is congested (i.e., CBR exceeds $CBR_{max}$) the *ThresholdBinary* value takes on a higher value, requiring high number of objects be uniquely detected by the ego vehicle to transmit the CPM.

### 3.2. CBR-Selective Scheme

Similar to the *CBR-binary* scheme, *CBR-selective* takes into account only resource availability. The difference is that *CBR-selective* selects CPM contents, i.e., a list of objects to include in a CPM based on the number of incoming CPMs that contain the same objects. It may decide to not transmit a CPM, if all the detected objects have been announced by a sufficient number of neighboring vehicles/infrastructure.

Similar to the *CBR-binary* scheme, upon reception of a list of objects detected by the ego vehicle, $A = [a_i]$ (from the *local perception module*), the *CPM content selection module* determines which objects to include in a CPM (see Figure 3).

As can be described in Algorithm 3, for each object, the module verifies, by referring to the columns (D) and (E) of the extended perception table if its neighbors (vehicle or RSU) have detected the same object. If the object, $a_i$, has been detected by more than *ThresholdSelective* neighbors, the object is removed from the list $A$, and not considered for inclusion in an outgoing CPM. Once the above verification procedure has been finalized and if the list $A$ is not empty, the objects contained in the list will be included in a CPM

and be transmitted by the CPM server module. Otherwise, the *CPM content selection module* fills a CPM with the remaining objects in the list and provides to the *CBM-server* for a transmission.

---

**Algorithm 3:** *CBR-selective* algorithm.

---

/* The objects that have been perceived only by the local sensors or
   the number of neighbors that detected the same object is below a
   threshold value, *Threshold*, will be included in the CPM. If there
   is no such an object, no CPM will be transmitted.              */

**Initialization:** ThresholdControl(*Init*, *InitThr*$_{selective}$, $\delta_{selective}$)

**Input**　　　: $A = [a_i]$, $i = 1, 2, \cdots, n$, a list of objects (more precisely, object IDs), perceived by the local perception module.

**Output**　　: $A$, the list of objects to include in an outgoing CPM; if the list is empty, CPM will not be transmitted.

*objectList*=CbrSelective($A$)

**if** *size*(*objectList*) $> 0$ **then**
　│ *Fill a CPM with the objects in objectList and provide the CPS server module for a*
　│ *transmission.*
**end**

**Function** CbrSelective($A$):
　│ *ThresholdSelective*=ThresholdControl(*GetThr*,0,0)
　│ **for** $i \leftarrow 0$ **to** $n-1$ **do**
　│　│ *numRedundance* = $Column\_D(a_i) + Column\_E(a_i)$
　│　│ **if** *numRedundance* > *ThresholdSelective* **then**
　│　│　│ *A.remove*($a_i$)　　// the object has been perceived by a higher
　│　│　│　than *Threshold* number of neighbors; remove the object from
　│　│　│　the list.
　│　│ **end**
　│ **end**
　│ **return** $A$
**End Function**

---

Same as the *CBR-Binary* scheme, *ThresholdSelective* is adjusted by the *CPM content selection module* by the threshold control algorithm detailed in Algorithm 2. The initial value of *Threshold* and the step parameter $\delta$ are set to predefined values *InitialThreshold*$_{selective}$ and $\delta_{selective}$.

As mentioned, *ThresholdSelective* takes on a lower value when the channel is sparse, and a higher value when the channel is congested. Therefore, in the *CBR-selective* scheme, higher level of redundancy is allowed if the channel is sparse and the redundancy is highly restricted if the channel is congested.

### 3.3. CBR & Infra-Selective Scheme

*CBR & Infra-selective* takes into account on radio resource availability but also presence of roadside infrastructure that is providing CP service. The motivation behind taking into account the presence of roadside infrastructure is that the infrastructure can be installed in such a way it has an extended (and often line-of-sight (LoS)) communication and sensor coverage [36]. Therefore, if an object has been already announced by the infrastructure, there is not much value to add by the ego vehicle with a transmission of a CPM containing this object. Indeed, in our previous paper [8], we showed that when CPMs are generated and transmitted by roadside infrastructure, vehicles in its communication coverage do not really need to send CPMs because the infrastructure alone can provide extended perception without needing much of radio resource.

Similar to the *CBR-selective*, upon reception of a list of objects detected by the ego vehicle, $A = [a_i]$ (from the *local perception module*), the *CPM content selection module* determines which objects to include in a CPM (see Figure 3). As shown in Algorithm 4, the *CPM content selection module* will first run the *CBR-selective* algorithm targeting only the column (D) of

the *extended perception table*. In other words, the algorithm verifies the level of redundancy of any object that is detected by the ego vehicle and neighboring vehicles, and it discards objects if this level of redundancy exceeds a threshold value *ThresholdSelective*. *Threshold-Selective* is permanently adapted based on the measured CBR by the threshold control Algorithm 2. The *CPM content selection module* will then further run *Infra-selective* algorithm and removes the objects from the list that have already been informed by the roadside infrastructure. Finally, if the resulting list is not empty, the module fills a CPM with the remaining objects in the list and provide the CPM to the *CPS server* for a transmission.

---

**Algorithm 4:** *CBR & Infra selective* algorithm.

---

```
/* Check if vehicle received a CPM from RSU. If not, record
   occurrence of each object received in CPM. Otherwise, record
   occurrence of each object received in CPM. If there is no such an
   object, no CPM will be transmitted.                              */
```
**Initialization:** ThresholdControl(*Init*, *InitThr$_{selective}$*, $\delta_{selective}$)

**Input**          : $A=[a_i]$, $i = 1, 2, \cdots, n$, a list of objects (more precisely, object IDs), perceived by the local perception module.

**Output**        : *objectList*, list of objects to include in an outgoing CPM; if objectList is empty, CPM will not be transmitted.

*objectList*=CbrInfraSelective($A$)

**if** *size(objectList)* $> 0$ **then**

  | *Fill a CPM with the objects in objectList and provide the CPS server module for a transmission.*

**end**

**Function** CbrInfraSelective($A$):

  | *ThresholdSelective*=ThresholdControl(*GetThr*,0,0)

  | **for** $i \leftarrow 0$ **to** $n - 1$ **do**

  |   | *numRedundance = Column_D($a_i$)* **if**

  |   | *numRedundance > ThresholdSelective* **then**

  |   |   | *A.remove($a_i$)*      // the object has been perceived by a higher
  |   |   | than *ThresholdSelective* number of neighboring vehicles;
  |   |   | remove the object from the list.

  |   | **end**

  | **end**

  | *objectList*=InfraSelective($A$)

  | **return** *objectList*

**End Function**

**Function** InfraSelective($A$):

  | **for** $i \leftarrow 0$ **to** $n - 1$ **do**

  |   | **if** *Column_E($a_i$)* $> 0$ **then**

  |   |   | *A.remove($a_i$)* // the object is detected by an infrastructure;
  |   |   | remove the object from the list.

  |   | **end**

  | **end**

  | **return** $A$

**End Function**

---

## 4. Performance Evaluation

In this section, proposed schemes are assessed by comparing their performances against the *Default* scheme via the following performance indicators:

- PDR (packet delivery ratio): indicates the reliability of V2X communications, measured for each pair of transmitter and receiver as the ratio between the number of received packets over the number of transmitted packets. Ideally, the proposed schemes should improve PDR, i.e., each transmitted packet is successfully received by the receiver by improving resource use.

- Awareness rate: indicates the knowledge an ego vehicle has about nearby road users. It is measured as the ratio between the number of road users detected based on different perception means (on-board sensors, CPM or fusion of both sources) over the total number of road users situated in a target vicinity of the ego vehicle. Because the *Default* scheme periodically transmits the complete list of the perceived objects by the sensors (with a sufficient confidence), it tends to provide high awareness rate. The awareness rate may degrade in *Default* scheme if the channel is too congested. The objective of the proposed schemes is, hence, to provide at least the same level of awareness rate as *Default* scheme.
- CBR (channel busy ratio): indicates the channel occupation, measured as the ratio of time during which channel is sensed as busy (the receive signal power exceeds a given threshold) over a given monitoring period. The objective of the proposed scheme is to reduce the CBR, while improving PDR and at least maintaining the awareness rate (of the *Default* scheme).

### 4.1. Simulation Setup

We conducted simulations using Veins simulator [37], which combines the SUMO traffic simulator [38] and the OMNeT++ network simulator [39]. As shown in our previous work [13], a significant performance degradation in CPM transmission can be observed in dense traffic conditions due to a large number of communication entities competing for the radio resource. Therefore, in this work, simulation study focuses on a dense traffic scenario in a 4-lanes highway road section with an average inter-vehicle distance of 20 m, i.e., a road density of 50 vehicles/km/lane. Then, we vary the penetration rate for of connected vehicles (i.e., those have CP server/client modules): 20%, 50%, 75% and 100% .

The ITS G5 technology is used as the V2X communication technology. All the vehicles embed on-board sensors. When a roadside unit (RSU) is available, it is equipped with sensors and ITS G5 technology. The perception models of the vehicle on-board and the roadside sensors are presented in [40]. These models have been designed to simulate detection of objects that are in the sensors' FoV. On the one hand, vehicles' on-board sensors model integrates an occlusion model, i.e., takes into account occlusion induced by the presence of other vehicles. On the other hand, roadside perception model does not integrate such an occlusion model because roadside sensors can be installed sufficiently high. Table 3 lists the simulation parameters, including sensor coverage, antenna height and communication range for both vehicles and RSUs.

Finally, schemes proposed in Section 3 are evaluated with default parameters presented in Table 3. First, we consider threshold control function for *CBR-binary* and *CBR-selective* schemes is active when CBR is between 0.6 and 0.7 (respectively $CBR_{min}$ and $CBR_{max}$ value) and initial threshold in the number of object per CPM, $InitThr_{binary}$, is set to 0 with a step, $\delta_{binary}$, of 0.1 for CBR-binary scheme as the initial number of redundancy, $InitThr_{selective}$, for CBR-selective scheme is set to 5 with a step, $\delta_{selective}$, of 1.

**Table 3.** Simulation parameters.

| Parameter | Value |
| --- | --- |
| Road length | 1000 m |
| Number of lanes | 4 |
| Number of directions | 2 |
| Inter-car distance | 20 m |
| Vehicle sensor coverage ($D_{max}^{v}$) | 100 m |
| RSU sensor coverage ($D_{max}^{RSU}$) | 150 m |
| RSU sensor FoV | 360° |
| Lane width ($w$) | 3 m |
| Vehicle length ($d_{len}$) | 5 m |
| Vehicle width ($d_w$) | 2 m |
| Data rate | 6 Mbps |
| RSU antenna height | 3 m |
| Vehicle antenna height | 1.5 m |
| Vehicle communication coverage | 400 m |
| RSU communication coverage | 800 m |
| ($InitThr_{binary}$, $\delta_{binary}$) | (0, 1) |
| ($InitThr_{selective}$, $\delta_{selective}$) | (5, 0.1) |
| ($CBR_{min}$, $CBR_{max}$) | (0.6, 0.7) |

*4.2. CPM Generation Control Taking into Account Radio Resource Availability*

In this subsection, the schemes that take into account radio resource availability, i.e., *CBR-binary* and *CBR-selective*, are evaluated. Figure 4 shows the PDR performances of the different schemes. As expected, the higher the penetration rate, the lower PDR is, especially for the *Default* scheme. When the penetration rate is greater than 0.75%, both the *CBR-binary* and *CBR-selective* schemes perform better than the *Default* scheme. This can be explained as in these conditions, the proposed algorithms can sense the degradation of channel resource availability and control CPM's transmission opportunity or contents. We notice that median PDR value can be maintained above 75% even in the highest-load scenario, i.e., 100% of penetration rate. This clearly indicates the benefit of the proposals in improving communication reliability.

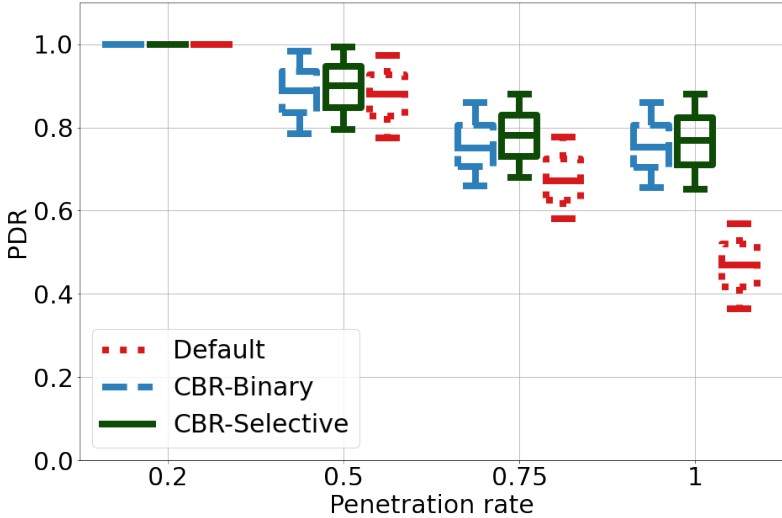

**Figure 4.** PDR function of penetration rate for CBR-Selective, CBR-Binary and Default algorithms.

Figure 5 illustrates the awareness rates of the three schemes. First, we can notice that regardless of the scheme, the awareness rate increases with the increase of the penetration rate. This is an expected behavior because, in low penetration rate scenario, vehicles

must rely mainly on their on-board sensors, which cover a small part of their surrounding environment, and hence they have low awareness rate. On the other hand, for high penetration rates, higher number of connected vehicles contribute in building collective perception, extending perception coverage, consequently increasing the awareness rate. Moreover, the figure confirms that *CBR-binary* and *CBR-selective* schemes have similar performances with the *Default* scheme.

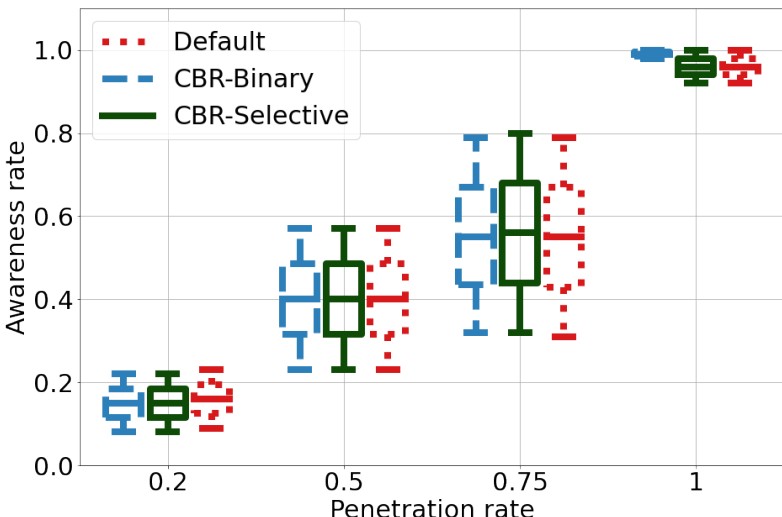

**Figure 5.** Awareness rate function of penetration rate for CBR-Selective, CBR-Binary and Default algorithms.

Figure 6 shows the CBR performances of the schemes. As the figure shows, in *CBR-binary* and *CBR-selective*, CBR remains below 0.6 even when the penetration rate is greater than 0.75%. The *CBR-selective* scheme tends to show even lower CBR, conceivably, because it reduces the CPM packet size.

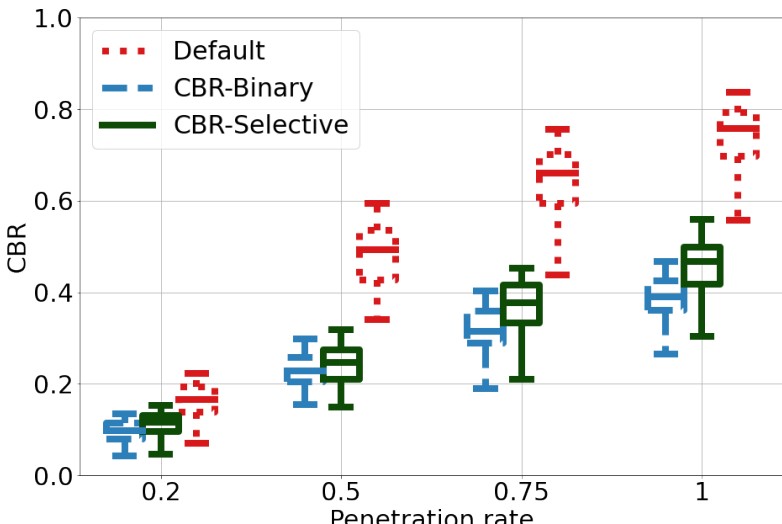

**Figure 6.** CBR function of penetration rate for CBR-Selective, CBR-Binary and Default algorithms.

The above results demonstrate that *CBR-binary* and *CBR-selective* schemes, by taking into account CBR, could successfully improve PDR and reduce radio resource use while maintaining the awareness rate of the *Default* scheme.

### 4.3. CPM Generation Control Taking into Account Both the Radio Resource and the Infrastructure Availability

The results of the previous subsection showed that it is possible to improve CPM performance by taking into account only CBR. However, communication performances in terms of PDR may not be satisfying to reach requirements of road safety applications [8], especially in dense conditions as shown in Figure 4. To overcome such limitations, *CBR & Infra selective* scheme considers two contexts: resource and infrastructure availability.

Figure 7 compares the PDR performance of the *CBR & Infra selective* and *Default* schemes. The figure clearly shows that PDR is nearly 100% in all the scenarios for *CBR & Infra selective* scheme, providing up to 60% of improvement in comparison to the *Default* scheme. Indeed, we have shown in our previous work [13] that the presence of a roadside infrastructure can positively impact the CPM performance. Therefore, as fewer entities are sending data packets when RSU takes over vehicles for CPM transmission, higher PDR is observed, indicating more reliable communication.

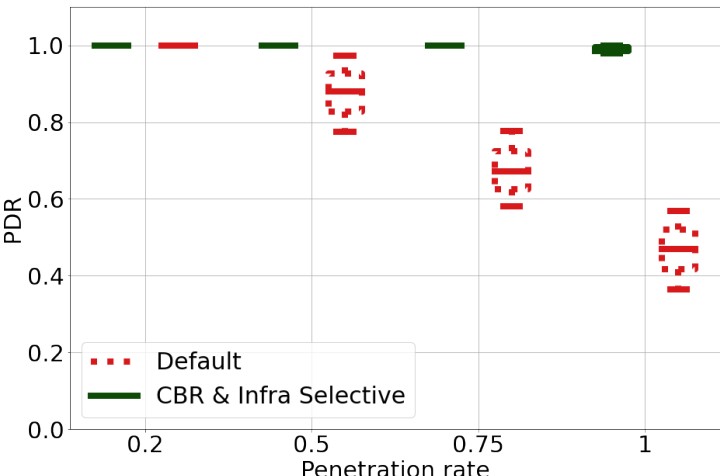

**Figure 7.** PDR function of penetration rate for Default, CBR & Infra selective algorithms.

Figure 8 shows awareness rate for the two schemes. As we can see the proposed approach is very similar to the performance obtained with the *Default* scheme. Consequently, collective perception performance are maintained with the proposed approach while communication performance are largely improved.

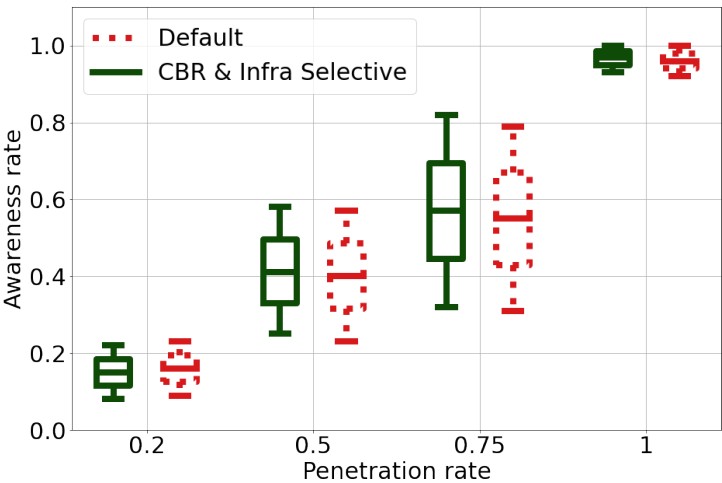

**Figure 8.** Awareness rate function of penetration rate for Default, CBR & Infra selective algorithms.

Finally, reduced channel usage is confirmed by Figure 9 showing that in the presence of a RSU, median CBR value remains below 40% which represents a decrease of 30% at 100% penetration rate for CPM transmission.

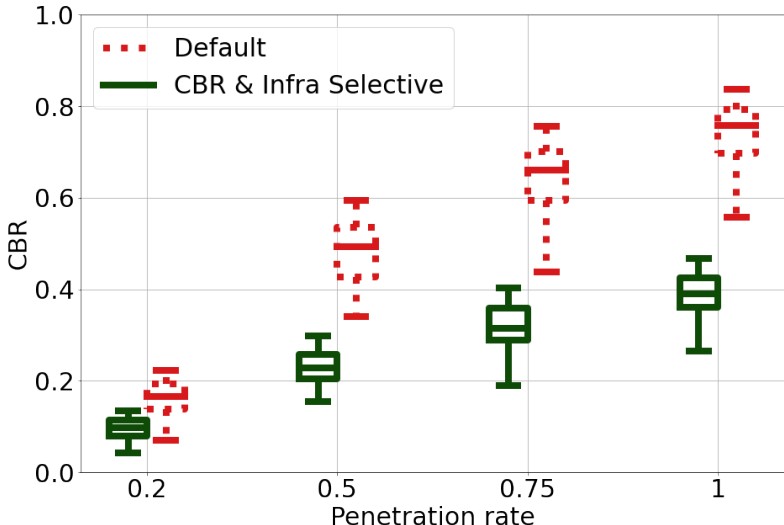

**Figure 9.** CBR function of penetration rate for Default, CBR & Infra selective algorithms.

To summarize, *CBR & Infra selective* scheme overcomes the *Default* algorithm and presents improved results in terms of radio resource use and packet delivery ratio. Indeed, the results confirm that solutions that control/select CPM contents tend to show improved performance. Furthermore, when a roadside infrastructure is available, relying on such an entity broadcasting CPMs has a major impact on improved resource use and PDR while still providing a sufficient level of situation awareness. This is because, in the proposed *CBR & Infra selective* scheme, vehicles remove contents from their CPM, if the same contents have been announced by an RSU, resulting in fewer and shorter CPM packets being transmitted in the channel and hence individual CPMs have larger informative quantity. Therefore, by adapting CPM transmission to the context, in particularly by replacing V2V communication by I2V communication seems to be greatly beneficial.

*4.4. Discussion*

Situation awareness and packet delivery ratio have been the main key performance indicators used in the literature to assess reliability of CPM generation approaches [29,30,32]. Thus, we believe they are the primary indicators for the evaluation of collective perception performances. Additionally, metrics such as payload size, CBR, detected object redundancy can complement the above-mentioned indicators for collective perception service evaluation [28,29].

Other researchers have demonstrated, similarly to our work, that awareness rate can be maintained at a high level while CPM generation is controlled from external factors to minimize impacts on the communication channel [29,30]. By studying multiple CPM generation and data content selection schemes, this work highlights major factors (or contexts) which influence reliability of collective perception service and how to take them into consideration for intelligent message dissemination.

First, it appears clearly from our results that controlling CBR is essential to maintain channel availability, thus, providing higher reliability. Second, when an infrastructure is available, it should take message dissemination over the vehicles as it benefits from better capacity in terms of communication and perception coverage. Therefore, *CBR & Infra selective* schemes outperform the other approaches presented in this paper, especially in dense scenarios.

Complementary, as sparse scenarios are less subject to degradation of communication reliability, a higher degree of data redundancy could be accepted. This would improve other performance indicators such as object localization accuracy. Indeed, in such scenarios, higher velocities of road users may degrade this indicator due to high dynamics of the environment. Moreover, evaluating object localization error is essential for the validation of automated systems, e.g., CAV, and has been considered in [30] where measurement errors are simulated. In our work, this aspect could not be evaluated; however, it will be relevant for future research especially when targeting real-world experimentation.

## 5. Conclusions

We presented context-aware communication schemes for CPM content selection and transmission control. Two types of contexts have been considered: radio resource and infrastructure availability. *CBR-binary* and *CBR-selective* schemes take into account radio resource availability. First, in *CBR-binary*, the CPM server transmits the whole list of objects if only if the number of objects that have been detected by only the ego vehicle exceeds a threshold, determined from the current channel busy ratio (CBR). Second, the *CBR-selective* scheme fills outgoing CPMs only with the objects that have been detected by less than a given number of neighbors, which is a threshold value determined from the current CBR. The simulation results show that in comparison to the *Default* scheme that does not control CPM contents (i.e., periodically broadcasts the complete list of the perceived objects), both the schemes, *CBR-binary* and *CBR-selective*, improve packet delivery ratio (PDR) and reduce CBR while offering the same level of awareness rate as the *Default* scheme. In particular, the results of *CBR-selective* show that it is possible to maintain the awareness rate with lower resource consumption (up to 30% of CBR reduction).

We also evaluated the performance of *CBR & Infra selective* scheme, which takes into account not only radio resource use but also if the objects have been announced by a roadside infrastructure. The performance improvement of the scheme is remarkable. In particular, it offers 100% of PDR regardless of the penetration rate of connected vehicle and reduces CBR by up to 30% while maintaining the same level of awareness rate as *Default*.

Our future work includes implementation of the schemes in a V2X test-bed platform and demonstration of its effectiveness by real-world experimentation.

**Author Contributions:** Investigation, A.C.; Methodology, P.M. and O.S.; Supervision, O.S.; Validation, A.C.; Visualization, A.C.; Writing—original draft, A.C., P.M. and O.S.; Writing—review & editing, P.M. and O.S. All authors have read and agreed to the published version of the manuscript.

**Funding:** This research received no external funding.

**Conflicts of Interest:** The authors declare no conflict of interest.

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
