# Peer review of "Context-Aware Content Selection and Message Generation for Collective Perception Services"

_electronics, doi:10.3390/electronics10202509_

Round 1
Reviewer 1 Report
In this work, the authors propose three techniques that can leverage the congestion of the wireless medium, the availability of roadside infrastructure, and the neighboring collective awareness in cooperative intelligent transportation systems, to control the generation, content selection, and transmission of Collective Perceptive Messages (CPMs). The goal is to reduce information redundancy and radio utilization. The performance of all three proposals is evaluated under simulations using SUMO and OMNeT++.
The work is, to the best of my knowledge, original and the article well organized. The authors should carefully review the article for missing articles (a, an, the..) and minor typos. For example:
- lines 39-40 "[...] confirmed that generation of CPM is frequent continuing small messages and so reporting few objects, hence, degrading [...]" -> confusing/malformed sentence, revise it.
- lines 72-73 "[...] (DCC) framework which is to allow communication [...]" -> "framework to allow [...]"
- line 171 "the architecture maybe applied" -> "may be applied"
- line 189 ", and" -> is the remaining of this item missing? otherwise delete ", and"
- line 206 "CPM if only if the number" -> "CPM if, and only if, the number"
- Algorithm 1, "If the number of such objects does not exceed a threshold" -> I believe it should be "does exceed" instead as the CPM is transmitted once the threshold is exceeded (from the algorithm description)
- lines 281 "CBR-seelctive" -> "CBR-selective"
- line 317 "the Sumo traffic" ->"the SUMO traffic"
The work fits the scope of the journal and is, in my opinion, mature enough for publication. However, the authors should consider the following minor points to improve the article:
- The problem statement is clear and the existing literature properly described. However, the authors should explicitly state which are the open issues that this work solves.
- Figures 2-9 are a bit small and should be enlarged for better readability. Additionally, all plots should adhere to the same style (either with open or with filled boxes).
- In lines 227-228 "It is up to each scheme to define the initial
228 value of Threshold and the value of d.". How should the user pick the right values? What practical considerations and tradeoffs should he have in mind when picking values for these parameters (e.g., awareness rate vs bandwidth use)? - Are there any Key Performance Indicators that could be used to compare the performance of these algorithms with other approaches found in the literature? It would be desirable to have a way to compare the performance of the proposed algorithms with other solutions already in the literature (besides the "default" approach, that is).
Reviewer 2 Report
In this paper, authors present several proposals about collective perception services.
The introduction provides a good perspective of the problem and there is a good related-work section .
Authors propose 3 different methods, two of them does not require the aid of RSUs to perform the selection of CPMs to be transmitted. The third method, makes use of the higher transmission capabilities of RSUs to perform that CPM selection.
The algorithms are well explained, and in general, the paper is easy to follow. Also, Authors provide a good evaluation section using simulations in high density scenarios. However, in my opinion, authors do not clarify whether is better to use which method. It is obvious that the method that makes use of RSUs is the one that better results obtains, but authors should discuss in which scenarios is better to use one method over the others. For example, in areas where there are few RSUs with a lower coverage, which of the other methods would be better to implement?
There are some minor typos, that should be checked. i.e, Section 3, second paragraph, “…the sensors. The vehicle also has collective perception service (CPS) consisting of a client module (CPS client), which processes received CPMs, and a server module (CPS client), which is responsible for generating and transmitting CPMs.” -> the server module should be (CPS Server) as shown in figure 3.
